# Dysphagia in Children, Do Not Blame Eosinophils Too Quickly

**DOI:** 10.3390/children10010063

**Published:** 2022-12-28

**Authors:** Antoine Debourdeau, Jean-Michel Gonzalez, Marc Barthet, Véronique Vitton

**Affiliations:** 1Gastroenterology and Endoscopy Unit, Hôpital Nord, Assistance Publique des Hôpitaux de Marseille, Aix Marseille Université, 13015 Marseille, France; 2Endoscopy Unit, CHU Montpellier, Montpellier University, 34000 Montpellier, France; 3Gastroenterology Unit, CHU de Nîmes, Montpellier University MUSE, 30000 Nîmes, France

**Keywords:** achalasia, eosinophilic esophagitis, esophageal manometry, esophageal motor disorder

## Abstract

Dysphagia in children is a relatively frequent symptom in childhood, and the main causes are congenital and linked to ear–nose–throat etiologies. However, non-congenital esophageal dysphagia is less common, and the main cause in such cases is eosinophilic esophagitis (EoE). When there is no response to a well-conducted treatment, with normalization of histology, the diagnosis of EoE must then be reconsidered. Here, we present the case of a 10-year-old patient whose initial diagnosis of eosinophilic esophagitis delayed the diagnosis of type III achalasia.

## 1. Case

A 10-year-old boy came to our unit for progressively worsening dysphagia evolving for 1 year. There was no personal or family medical history, including no asthma or atopic eczema. The patient reported regurgitation and retrosternal pain without a real impaction episode.

A first upper endoscopy was performed, which showed no stricture, and whose esophageal biopsies evidenced an infiltration of the esophageal wall with >15 eosinophils per field of optical microscope.

A first line of treatment with esomeprazole 40 mg was then conducted for 2 months, with no improvement in dysphagia. A new upper endoscopy was performed, the biopsies of which evidenced a persistence of eosinophilic mucosal infiltrate (27 eosinophils per high-powered microscopic field). A topical steroid treatment (viscous solution of budesonide) was then started.

Despite this management, the patient’s general condition worsened, with major weight loss (up to 20% of weight) and school dropout. As dysphagia persisted and worsened, the patient had to be fed by enteral nutrition through a nasogastric tube.

A new gastroscopy was performed four months after the start of steroid therapy, which revealed esophageal candidiasis, thought to be related to the topical steroid. Esophageal biopsies showed a complete histological response with <5 eosinophils per field.

Three months later, given the persistence of dysphagia after effective treatment of esophageal candidiasis, an esophagogram with a barium swallow examination was performed (Figure 1), highlighting an insufficient esophageal emptying associated with esophageal dilatation and no relaxation of the lower esophageal sphincter, suggestive of achalasia.

An esophageal manometry was then performed (Figure 2), which showed an elevated integrated relaxation pressure of the lower esophageal sphincter (44.3 mmHg) associated with the presence of esophageal spasms, leading to the diagnosis of type III esophageal achalasia.

Investigations did not reveal corticotropic insufficiency nor alacrimia that could be indicative of a triple A syndrome.

The patient was then treated with per oral endoscopic myotomy (POEM) with tailored myotomy based on the length of the spasms on the manometry. The procedure had an immediate effect on dysphagia, with no recurrence of symptoms at clinical follow-up. The barium swallowing was improved, with no stagnation above the cardia 6 months after the procedure, an improvement in his general condition, and a return to school.

A final follow-up endoscopy was made 6 months after the POEM, and showed no eosinophils on esophageal biopsies. The patient had stopped the budesonide treatment since the myotomy.

## 2. Discussion

Dysphagia in children is a relatively frequent symptom in childhood, and approximately 1% of children will experience swallowing difficulties, the main causes of which are linked to ENT (Ear–Nose–Throat) issues and associated with a particular background in children with cerebral palsy, traumatic brain injury, and airway malformation [1,2]. However, non-congenital esophageal dysphagia is less common in children, and eosinophilic esophagitis (EoE) is the main cause in such cases. Symptoms of eosinophilic esophagitis vary according to the age of the patients, and dysphagia is more frequent after 10 years, with 28% of children suffering from it at a median age of 13 years, while young children usually experience vomiting, regurgitation, abdominal pain, and feeding disorders [3]. On the contrary, dysphagia and food impaction are rather predominant in adolescents and adults, and are generally related to advanced tissue remodeling.

Upper endoscopy is normal in 17% of cases of eosinophilic esophagitis, stricture is present in 21% of cases, and a narrow-caliber esophagus occurs in 9% of cases [4].

In EoE, esophageal biopsy is the gold standard for diagnosis, but this does not explore tissue remodeling or fibrosis, nor does it provide information regarding the presence of esophageal fibrosis and narrowing. However, dysphagia in eosinophilic esophagitis is most often related to fibrous narrowing of the lumen, and not to the inflammatory activity of eosinophils. Strictures in this setting are usually focal and located in the upper 2/3 of the esophagus. Esophageal endoscopy is imprecise, and can be misleading because it does not recognize narrowing between 11 mm and the diameter of the scope (7–9 mm), whereas a barium esophagogram shows this narrowing in 71% of adults [4] and 55% of the children [5]. In our case, the persistence of eosinophils of infiltration with a worsened dysphagia should have raise the question of the presence of esophageal stricture linked to EoE, especially in the upper esophagus.

In the case of eosinophilic esophagitis, the lack of clinical improvement after two lines of treatment (proton pump inhibitors and a topical steroid) is very unusual and occurs in only 5% of cases [6]. In such cases, the lack of clinical response in the presence of a histological response should question the diagnosis of eosinophilic esophagitis, and other associated diagnoses should be considered as dysphagia etiologies. In our case, the patient had three lines of therapeutics: PPIs, topical steroid, and, finally, an elimination diet with total enteral nutrition.

Achalasia is a rare disease, the incidence of which is rarer in the pediatric population than in adults (0.18/100,000 children per year) [7].

Achalasia in children develops at an average age of 14 years, with a majority of cases being idiopathic [8]. Some may be related to genetic syndromes, such as triple A syndrome, characterized by the triad of achalasia, alacrimia, and adrenocorticotropic hormone (ACTH)-resistant adrenal insufficiency [9]. Nevertheless, the diagnosis of genetic forms is often made earlier in childhood, before the age of 10, and the response to treatment is poorer in these cases [8,9].

The diagnosis of achalasia is suspected on the esophagogram, and the subtype of achalasia is confirmed by high-resolution esophageal manometry.

Pneumatic balloon dilation is recommended as a first line of treatment for type I and II achalasia, while a POEM is recommended for type III achalasia (which allows treatment by myotomy at of the esophagus body the lower esophageal sphincter) [10].

Although there are no data in the literature to support this impression, we assume that esophageal stasis related to achalasia was responsible for the increase in mucosal eosinophils count, since this disappeared after POEM when the patient had stopped budesonide for 6 months.

The association of two rare esophageal pathologies can be possible, and this case highlights the key importance of esophagograms and manometry in the diagnosis of dysphagia when endoscopy shows no abnormality. In addition, these approaches are crucial in the case of weight loss, as it has been shown that esophageal motor disorders are more frequently found in cases of weight loss associated with dysphagia in children [11]. In EoE, if there is no improvement in dysphagia despite a histological response, an evaluation by barium and esophageal manometry should be considered.

## Figures and Tables

**Figure 1 children-10-00063-f001:**
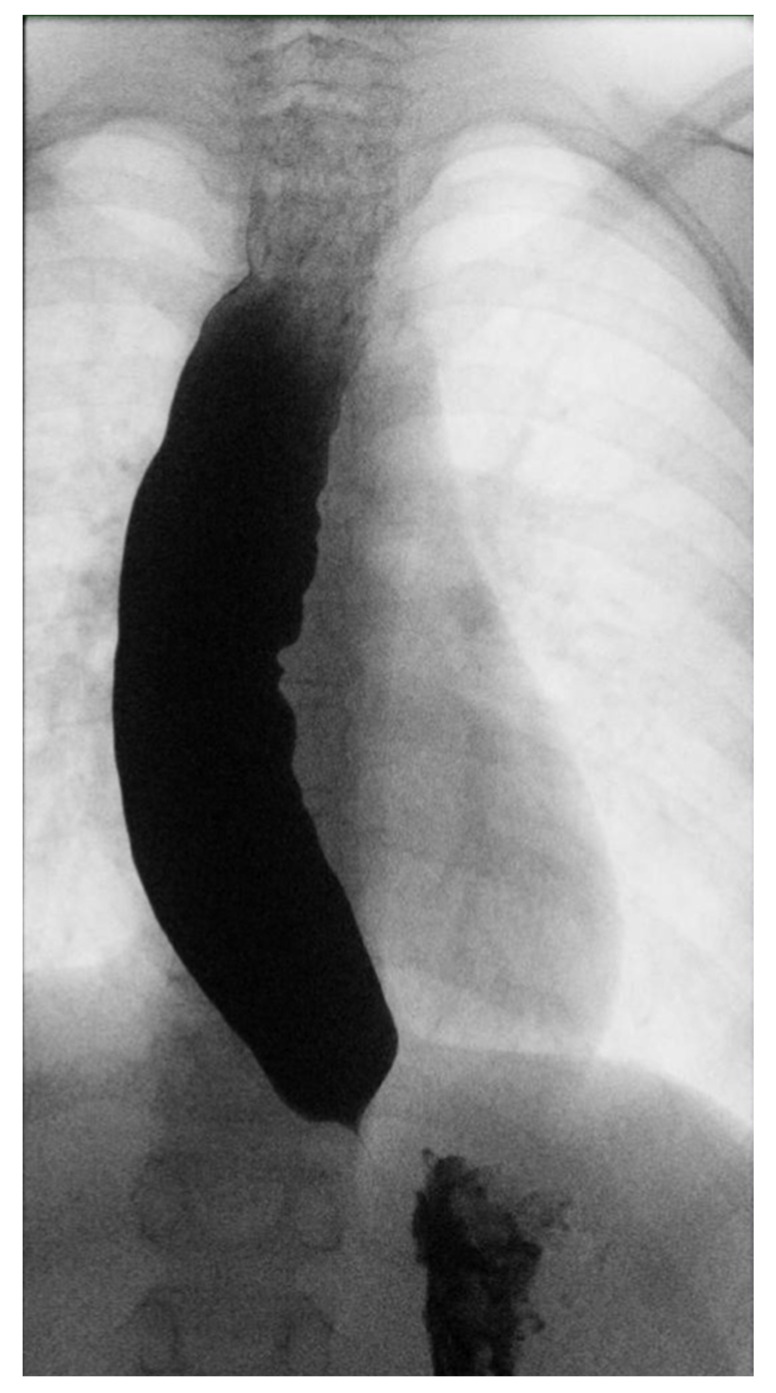
Esophageal barium swallow.

**Figure 2 children-10-00063-f002:**
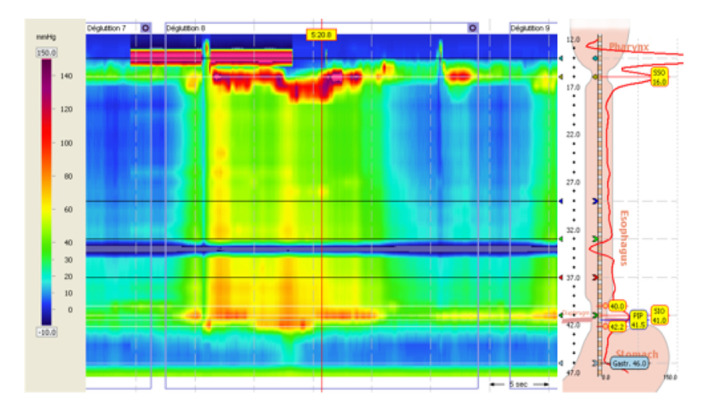
Esophageal high-resolution manometry.

## Data Availability

It’s not applicable.

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
