# Peer review of "Dysphagia in Children, Do Not Blame Eosinophils Too Quickly"

_children, 2022, doi:10.3390/children10010063_

Round 1
Reviewer 1 Report
Dear Authors,
your case report is intersting but there is need to improve discussion:
1) in case description (line 55) need to specify how do you perform follow up: is it clinical? or did you perform endoscopy / manometry/ barium swallow?
2) line 58: what do you mean with ENT? explain
3) line 61: modify EEo with EoE abbreviation
4) line 63: symptoms vary from age must be better explained. Young children and toddlers usually experience vomiting, regurgitation, abdominal pain, and feeding refusal. On the contrary, dysphagia and food impaction are predominant in adolescents and adults, and are generally related to advanced tissue remodeling
5)line 68-72: you are discussing narrowing of the esophagus seen in endoscopy. Since esophageal biopsy through upper gastrointestinal endoscopy still remains the gold standard diagnostic test to perform when EoE is suspected, you must discuss on this point and how did you applied it in your case.
6) at line 73 you discuss about two lines of treatment, while according to the last consensus guidelines, proton pump inhibitors (PPIs) are considered first-line therapeutic options on the same level as steroids and elimination diet. why don't you consider dietary treatment?
7) all the last paragraph from line 90 to the end must be modified because EoE diagnosis must be questioned also according to histology, not only endoscopy. Moreover you should clarify what could be the right diagnostic protocol in this cases: would you suggest performing manometry and barium swallow before histology?
8) figure 3 is not clear and maybe superfluous.
Author Response
Dear reviewer,
We want to thank you for your comments that had helped us to clarify and improve the manuscript.
You can see below a point-by-point response to your comments.
Sincerely,
Dr Antoine DEBOURDEAU
1) in case description (line 55) need to specify how do you perform follow up: is it clinical? or did you perform endoscopy / manometry/ barium swallow?
Thank you for your comments, we precised that in he manuscript.
2) line 58: what do you mean with ENT? explain
ENT is the abbreviation of Ear-Nose-Throat. I mentioned that in the manuscript
3) line 61: modify EEo with EoE abbreviation
Done
4) line 63: symptoms vary from age must be better explained. Young children and toddlers usually experience vomiting, regurgitation, abdominal pain, and feeding refusal. On the contrary, dysphagia and food impaction are predominant in adolescents and adults, and are generally related to advanced tissue remodeling
We modified the manuscript thank to your suggestions.
5)line 68-72: you are discussing narrowing of the esophagus seen in endoscopy. Since esophageal biopsy through upper gastrointestinal endoscopy still remains the gold standard diagnostic test to perform when EoE is suspected, you must discuss on this point and how did you applied it in your case.
we modified the text with this : "
In EoE, esophageal biopsy is the gold standard for diagnosis, but this does not explore tissue remodeling and fibrosis, and does not give information on the presence of esophageal fibrosis and narrowing. However, dysphagia in eosinophilic esophagitis is most often related to fibrous narrowing of the lumen, and not to the inflammatory activity of eosinophils." line 71-76 " In our case, the persistence of eosinophils of infiltration with a worsened dysphagia should have raise the question of the presence of esophageal stricture linked to EoE, especially in the upper esophagus." lines 80-826) at line 73 you discuss about two lines of treatment, while according to the last consensus guidelines, proton pump inhibitors (PPIs) are considered first-line therapeutic options on the same level as steroids and elimination diet. why don't you consider dietary treatment ?
You are right, dietary treatment is a good option, and is considered as a first line treatment. Nevertheless, in our experience, we have poor adherence to this solution because of the changes it causes in the life habits of our patients. In this specific case, dietary management was done because the patient had total enteral feeding with a nasogastric tube.
"As dysphagia persisted and worsened, the patient had to be fed by enteral nutrition on a nasogastric tube." line 39-407) all the last paragraph from line 90 to the end must be modified because EoE diagnosis must be questioned also according to histology, not only endoscopy. Moreover you should clarify what could be the right diagnostic protocol in this cases: would you suggest performing manometry and barium swallow before histology?
In our case, we had a dissociated response between the histological response (line 42-43) and a lack of improvement of the dysphagia.
the message we want to transmit is the following: "in case of no improvement of dysphagia despite a histological response, an evaluation by barium and esophageal manometry should be considered". we daddies this one 105-107 in place of the previous final sentence.
8) figure 3 is not clear and maybe superfluous.
Thanks, we deleted it.
Reviewer 2 Report
The case report "Dysphagia in children, don't blame eosinophils too quickly" presents a case of a 10-year-old boy diagnosed with eosinophilic esophagitis and then esophageal achalasia. Current literature cannot determine which is the primary condition and which is the result. It is important to publish more studies and clinical cases that may have an impact on understanding the pathogenesis of the coincidence of both diseases.
1. Title:
In my opinion, contractions should be avoided.
2. Abstract:
The abbreviation "ENT" is not expanded.
The abbreviation EEo is a rare abbreviation for eosinophilic esophagitis (EoE predominates). To standardize the nomenclature, I suggest using the abbreviations EoE.
Please correct the language error on line 14.
3. Case :
Was an upper gastrointestinal endoscopy performed between switching from PPI to topical steroid? If yes, what was the amount of eosinophils in hpf? If not, why?
How long has budesonide been used for treatment? After what time was the control endoscopy performed?
Line 47: no space "44.3mmHg"
Was EoE anti-inflammatory treatment still used after POEM? Did EoE symptoms resolve completely after POEM? Was endoscopy performed to monitor histological remission after clinical remission?
Line 52: should be Figure not figure
4. Discussion:
Line 57: Dysphagia is capitalized. Why?
Line 62-64: I suggest also citing more recent data on dysphagia in children with EoE. Suggested reference: doi: 10.3390/jcm9123869
Line 62: Please use the abbreviation instead of the full name eosinophilic esophagitis.
Line 73: Proton is capitalized. Why?
Line 72-76: Please cite the article this data comes from.
The discussion lacks information on the pathogenesis of both diseases. According to the authors, did EoE lead to achalasia? Could it be that achalasia was the cause of the increased amount of eosinophils in the esophagus? Or maybe both of these entities coexist independently of each other?
Author Response
Dear Reviewer,
We want to thank you for your work and your comments that helped us to improve the quality of this case report. We clarified it according your suggestions.
You ca see a pointt-by-point answer below.
Sincerely
Dr Antoine Debourdeau
- Title:
In my opinion, contractions should be avoided. --> We replaced don't by do not.
- Abstract:
The abbreviation "ENT" is not expanded. --> We replaced it by "Ear-nose-throat etiologies"
The abbreviation EEo is a rare abbreviation for eosinophilic esophagitis (EoE predominates). To standardize the nomenclature, I suggest using the abbreviations EoE. --> We replaced all the Exo by EoE.
Please correct the language error on line 14.
- Case :
Was an upper gastrointestinal endoscopy performed between switching from PPI to topical steroid? If yes, what was the amount of eosinophils in hpf? If not, why?
--> Yes and it evidenced Eo per hpf > 15, we mentioned it line 37-38.
How long has budesonide been used for treatment? After what time was the control endoscopy performed?
Endoscopy was made 4 months after budesonide start and budesonide was extended until the POEM intervention.
Line 47: no space "44.3mmHg" --> corrected
Was EoE anti-inflammatory treatment still used after POEM? Did EoE symptoms resolve completely after POEM? Was endoscopy performed to monitor histological remission after clinical remission?
the budesonide was discontinuantes after the POEM procedure and the upper endoscopy made 6 months after showed no more eosinophilic infiltration on biopsies.
Line 52: should be Figure not figure - corrected
- Discussion:
Line 57: Dysphagia is capitalized. Why? --> corrected
Line 62-64: I suggest also citing more recent data on dysphagia in children with EoE. Suggested reference: doi: 10.3390/jcm9123869
we inserted this citation line 69
Line 62: Please use the abbreviation instead of the full name eosinophilic esophagitis.
Line 73: Proton is capitalized. Why? corrected
Line 72-76: Please cite the article this data comes from.
10.1016/j.cgh.2020.01.024 we added this reference, it was a mistake from us. And we precise that "the lack of clinical improvement after two lines only occurs in 5%"
The discussion lacks information on the pathogenesis of both diseases. According to the authors, did EoE lead to achalasia? Could it be that achalasia was the cause of the increased amount of eosinophils in the esophagus? Or maybe both of these entities coexist independently of each other?
You are right, and we added a little paragraph to explain what we think about this :
"Although there are no data in the literature to support this impression, we assume that esophageal stasis related to achalasia was responsible for the increase in mucosal eosinophils count since this disappeared after POEM when the patient had stopped budesonide for 6 months." line 110-113
Round 2
Reviewer 2 Report
All concerns have been addressed.